# NH_2_-MIL-53(Al) Polymer Monolithic Column for In-Tube Solid-Phase Microextraction Combined with UHPLC-MS/MS for Detection of Trace Sulfonamides in Food Samples

**DOI:** 10.3390/molecules25040897

**Published:** 2020-02-18

**Authors:** Qian-Chun Zhang, Guang-Ping Xia, Jun-Yi Liang, Xiao-Lan Zhang, Li Jiang, Yu-Guo Zheng, Xing-Yi Wang

**Affiliations:** School of Biology and Chemistry, Key Laboratory of Chemical Synthesis and Environmental Pollution Control-Remediation Technology of Guizhou Province, Xingyi Normal University for Nationalities, Xingyi 562400, China; xiaguangping@xynun.edu.cn (G.-P.X.); liangjunyi@xynun.edu.cn (J.-Y.L.); zhangxiaolan@xynun.edu.cn (X.-L.Z.); jiangli@xynun.edu.cn (L.J.); zhengyuguo@xynun.edu.cn (Y.-G.Z.)

**Keywords:** monolithic capillary column, metal-organic framework, sulfonamides, solid-phase microextraction, ultra-high performance liquid chromatography-tandem mass spectrometry

## Abstract

In this study, a novel monolithic capillary column based on a NH_2_-MIL-53(Al) metal–organic framework (MOF) incorporated in poly (3-acrylamidophenylboronic acid/methacrylic acid-co-ethylene glycol dimethacrylate) (poly (AAPBA/MAA-co-EGDMA)) was prepared using an in situ polymerization method. The characteristics of the MOF-polymer monolithic column were investigated by scanning electron microscopy, Fourier-transform infrared spectroscopy, X-ray photoelectron spectroscopy, X-ray diffractometry, Brunauer-Emmett-Teller analysis, and thermogravimetric analysis. The prepared MOF-polymer monolithic column showed good permeability, high extraction efficiency, chemical stability, and good reproducibility. The MOF-polymer monolithic column was used for in-tube solid-phase microextraction (SPME) to efficiently adsorb trace sulfonamides from food samples. A novel method combining MOF-polymer-monolithic-column-based SPME with ultra-high performance liquid chromatography-tandem mass spectrometry (UHPLC-MS/MS) was successfully developed. The linear range was from 0.015 to 25.0 µg/L, with low limits of detection of 1.3–4.7 ng/L and relative standard deviations (RSDs) of < 6.1%. Eight trace sulfonamides in fish and chicken samples were determined, with recoveries of the eight analytes ranging from 85.7% to 113% and acceptable RSDs of < 7.3%. These results demonstrate that the novel MOF-polymer-monolithic-column-based SPME coupled with UHPLC-MS/MS is a highly sensitive, practical, and convenient method for monitoring trace sulfonamides in food samples previously extracted with an adequate solvent.

## 1. Introduction

Sulfonamides (SAs) are the most widely used antimicrobial veterinary drugs due to their low cost and high efficacy for targeting bacterial infections [1]. Nevertheless, SAs cannot be completely metabolized by animals, resulting in SAs residues from veterinary drugs entering the human body through the food chain, potentially posing many risks to consumers [2]. If SAs accumulate long-term, serious side effects related to urination and hematopoietic disorders may lead to further allergies, endocrine disorders, and digestive issues [3,4,5,6]. Thus, the investigation of SAs residues in animals and the potential health risks have attracted increasing attention. At present, in order to protect consumer safety, many countries and regions have established standardized acceptable levels for SAs in food products of animal origin. For instance, the European Union (EU) has adopted a maximum residue limit of 100 µg/kg for SAs in foods of animal origin [6,7,8]. It is, therefore, important to establish an effective, rapid, simple, and sensitive method for SAs determination for pre-enrichment before analysis. It is noteworthy to recognize that the sample matrices involved in the detection of trace analytes from foods are very complex. Thus, to reduce any possible interference from the sample matrix, sample pretreatment is a crucial step before UHPLC-MS/MS analysis can be performed [9].

Currently, various sample preparation techniques are employed to detect trace SAs in different matrices, including liquid-liquid extraction [10], dispersive liquid-liquid microextraction [11], solid-liquid extraction [12], matrix solid-phase dispersion [13], liquid membrane extraction [14] solid-phase extraction [15], and solid-phase microextraction (SPME). Among these methods, the main drawbacks of traditional fiber SPME are ease of breakage, low sorption capacity, and limited lifetime [16,17]. Therefore, monolithic capillary columns that incorporate polymers to replace the traditional fibers for SPME have been employed. Monolithic capillary SPME columns are easy to use with on-line systems to improve enrichment capacity and analysis sensitivity [17].

The core of SPME is adsorbent, which decides the extraction performance; an ideal SPME adsorbent has the characteristics of inherent porous structures, large specific surface area, adequate chemical stability, high adsorption capacity, etc. The use of boronate-affinity polymers usually imparts low back pressure, sufficient water stability, fast convective mass transfer, and resistance to both acidic and alkaline media [18,19,20]. However, the improvement of extraction performance is still worth researching. Metal–organic frameworks (MOFs) are materials that exhibit excellent characteristics, such as large specific surface areas, inherent porous structures, and high adsorption capacities [21,22,23]. MOFs have been applied for sample pretreatment and yielded exciting results; however, its use is limited because of inadequate chemical stability in moisture and inconvenient retrieval from the sample matrix [24,25,26].

A number of analytical methods, such as gas chromatography (GC) [27], liquid chromatography (HPLC) [28,29,30], and ultra-high performance liquid chromatography (UHPLC) combined with different detectors have been used for the detection and analysis of SAs residues. Compared to these methods, ultra-high performance liquid chromatography-tandem mass spectrometry (UHPLC-MS/MS) provides higher separation efficiency, selectivity, and sensitivity, making it the most commonly used approach for the analysis of trace SAs residues [31,32].

In this study, a novel porous monolithic capillary column is prepared via in situ polymerization to obtain a hybrid MOF-polymer monolithic column that combines the advantages of MOFs and SPME, which is used for the enrichment of eight SAs. Compared with the polymer monolithic column, the MOF-polymer monolithic column noticeably improves the extraction efficiencies of SAs, while the combination of liquid extraction, MOF-polymer-monolithic-column-based SPME with UHPLC-MS/MS provides an effective method for the detection of SAs in chicken and fish samples.

## 2. Results and Discussion

### 2.1. Characteristics of the MOF-Polymer Monolithic Column

Fourier-transform infrared (FT-IR) spectroscopy is a common method for characterization of chemical bonds and functional groups, providing insight into chemical groups that have been transformed or are present for a given compound. Figure 1A shows the FT-IR spectra of the MOF NH_2_-MIL-53(Al), the polymer only monolithic column, and the MOF-polymer monolithic column (see Section 3 for details regarding their preparation). For NH_2_-MIL-53(Al), the absorption bands at 3385 cm^−1^ and 3500 cm^−1^ correspond to the stretching vibration of -N-H bonds, which is attributed to the amino groups of NH_2_-BDC in the pores. The characteristic intense peak for C=O at 1670 cm^−1^ arises from C=O coordinated to Al [24,26,33]. For the polymer and MOF-polymer, the two sharp bands at 3562 cm^−1^ and 3497 cm^−1^ correspond to the stretching vibration of -O-H bonds, respectively, attributed to the presence of hydroxyl groups from methacrylic acid (MAA) and 3-acrylamidophenylboronic acid (AAPBA). The bands observed for the polymer and MOF-polymer at 2988 cm^−1^ and 2989 cm^−1^, respectively, are indicative of -C-H stretching, while those at 1729 cm^−1^ and 1730 cm^−1^, respectively, can be assigned to C=O stretching. The bands observed at 1390 cm^−1^ and 1391 cm^−1^ arise from the B-O stretching vibration of the polymer and MOF-polymer, respectively. This demonstrates that both the polymer and MOF-polymer possess the same characteristic peaks, whereas NH_2_-MIL-53(Al) differs. This is consistent with the different reactants used to generate the MOF-polymer monolithic column, and transformation of functional groups, ultimately resulting in the MOF-polymer possessing more functional groups similar to the polymer species.

X-ray photoelectron spectroscopy (XPS) was used to further analyze the surface chemical elemental composition of the MOF-polymer monolithic column. The full scan XPS spectrum of the MOF-polymer is presented in Figure 1B, displaying five characteristic peaks corresponding to Al 2p, B 1s, C 1s, N 1s, and O 1s. Furthermore, in the high-resolution spectra (Figure 1C,D). The N 1s XPS peaks from the MOF-polymer monolithic column (Figure 1C) can be resolved as two dominant peaks at 399.1 eV and 399.8 eV, corresponding to C_6_H_5_-NH_2_ and -C(O)-NH-, respectively. The O 1*s* peak (Figure 1D) can also be deconvoluted to peaks at 531.6 eV and 533.4 eV, reflecting the binding energies characteristic of -C(O)-NH- and -C(O)-O- bonds, respectively. These results further indicate that NH_2_-MIL-53(Al) was successfully incorporated in the MOF-polymer.

Figure 2A depicts the X-ray diffractometry (XRD) diffraction patterns of the MOF-polymer and pure NH_2_-MIL-53(Al). The peaks at 2*θ* = 9.3° and 18.2° are the main characteristic peaks of NH_2_-MIL-53(Al), with a minor peak at 2*θ* = 10.1° corresponding to trapped unreacted 2-amino terephthalic acid; these results are consistent with a previous report by Abedini et al. [33], showing an excellent match between the filler crystals and the polymer. The thermal stability of the MOF-polymer was also studied using thermogravimetric analysis (TGA). Figure 2B shows that the mass % gradually decreases until thermal degradation noticeably occurs at 350 °C, with the largest loss of mass occurring at 406.2 °C for the MOF-polymer. Therefore, the MOF-polymer is stable from room temperature to 120 °C.

The morphological structures of the MOF-polymer monolithic column and polymer monolithic column were investigated with scanning electron microscopy (SEM), as shown in Figure 3. The polymer monolithic column surface and pore structures are noticeably loose and porous (Figure 3A,B), whereas the homogeneous hybrid MOF-polymer monolithic column structure possesses favorable permeability (Figure 3C) and appears to consist of larger clustered units and fewer pores (Figure 3D,E). The morphological structure of the MOF-polymer monolithic column enhances the adsorption of SAs. Energy dispersive X-ray spectroscopy (EDS) was used to identify the major elements present in the MOF-polymer as B, C, N, O, and Al (Figure 3F). In addition, the surface area of the MOF-polymer monolithic column is larger (4.74 m^2^/g) than that of the polymer monolithic column (3.61 m^2^/g). These results are consistent with a porous, compact structure, which results in the larger surface area observed for the MOF-polymer monolithic column, which is essential to facilitate the efficient adsorption of analytes, thus improving the extraction of SAs.

### 2.2. UHPLC Procedure

These mobile phase compositions were considered as an important parameter between analyte separation and efficiency retention time. The effect of different mobile phase composition on the separation of the SAs with acetonitrile/water containing 0.5% acetic acid (*v*/*v*) and methanol/water containing 0.5% acetic acid (*v*/*v*) were studied at a constant flow rate of 0.3500 mL/min. Under optimal conditions, the gradient elution program involved an increase of the organic phase concentration from 10% to 40%. Figure 4A indicated that acetonitrile/water (2:8, *v*/*v*) containing 0.5% acetic acid (*v*/*v*) gave less separation time and better separation degree.

The effect of column temperatures has been investigated, including 36 °C and 46 °C in the separation of SAs. As shown in Figure 4B, with an increase of the column temperature, the results indicated the retention times slightly decreased. However, the temperature of 46 °C will reduce the lifetime of column, which can be attributed to macromolecule degradations. The column temperature of 36 °C was used.

### 2.3. Optimization of Extraction Conditions

To evaluate the optimal extraction conditions of the MOF-polymer monolithic column for SAs, several experimental parameters that influence the performance were optimized, including pH, extraction flow rate, desorption solvent, and desorption volume. These conditions were investigated using 5.00 mL of 50.0 µg/L standard solutions of the eight different SAs in ultra-pure water.

#### 2.3.1. Optimization of pH

The pH of the sample solution plays an important role in the amount of analyte extracted. Optimization of the pH was investigated from pH 2.0 to 10.0. As shown in Appendix A, excellent extraction efficiencies were achieved for all SAs at pH = 5.0. Therefore, the pH was maintained at 5.0 for subsequent experiments.

#### 2.3.2. Optimization of Extraction Flow Rate

The extraction flow rate was optimized from 100 to 200 μL/min. As shown in Appendix A, increasing the flow rate had no significant effect on the total amount of SAs extracted. Although slightly more SAs were extracted at a flow rate of 100 μL/min, a flow rate of 200 μL/min shortens the experimental time while still providing excellent sensitivity. Therefore, considering together the analytical time, extraction efficiency, moderate backpressure, and material stability, a flow rate of 200 μL/min was selected for further experiments.

#### 2.3.3. Optimization of Clean Volume and Desorption Solvent

To investigate the effective desorption of the eight SAs from the MOF-polymer monolithic column after extraction, the ultra-pure water volume used to clean the column was increased from 200 µL to 400 µL. As shown in Appendix A, increasing the clean volume from 200 µL to 400 µL is less effective for the removal of residual extraction solution from the MOF-polymer monolithic column. Therefore, 200 µL was chosen to clean the column. Different compositions of desorption solvent mixtures were then studied using five different proportions of solvents as acetonitrile/water (1:9, *v*/*v*), acetonitrile/water (2:8, *v*/*v*), acetonitrile/water (3:7, *v*/*v*), acetonitrile/water (1:9, *v*/*v*) containing 0.5% acetic acid (*v*/*v*), and acetonitrile/water (2:8, *v*/*v*) containing 0.5% acetic acid (*v*/*v*). Appendix A indicates that acetonitrile/water (2:8, *v*/*v*) containing 0.5% acetic acid (*v*/*v*) exhibited the best desorption efficiency for the eight SAs, and therefore this formulation was selected as the desorption solvent for subsequent experiments.

#### 2.2.4. Optimization of Desorption Flow Rate and Desorption Volume

The desorption flow rate was investigated between 100 and 200 μL/min, as shown in Appendix A. With increasing desorption flow rate using acetonitrile/water (2:8, *v*/*v*) containing 0.5% acetic acid (*v*/*v*), the desorption efficiency slightly decreased. Therefore, 100 μL/min was selected as the desorption flow rate, and while keeping this constant, the desorption volume was then investigated using 200 µL to 500 µL of acetonitrile/water (2:8, *v*/*v*) containing 0.5% acetic acid (*v*/*v*). Appendix A shows that there is no significant effect upon changing the desorption volume from 200 µL to 400 µL for most of the SAs; 400 µL was, therefore, chosen as the desorption volume.

### 2.4. Adsorption Characteristics of the MOF-Polymer Monolithic Column

#### 2.4.1. Extraction Capability and Adsorption Mechanism

Using the collective above optimized conditions, static isotherm adsorption experiments were performed using different concentrations of SAs to further explore the MOF-polymer monolithic column adsorption process. As shown in Figure 5A, with increasing initial extraction concentration, the MOF-polymer monolithic column adsorption capacity increases for all SAs. When Sulfadiazine (SDZ), sulfathiazole (STZ), sulfamerazine (SMI), sulfamethazine (SMZ), sulfamonomethoxine (SMM), sulfamethoxazole (SMX), sulfisoxazole (SIZ), and sulfadimethoxine (SDM) extraction concentrations were 1.6 mg/L, 2.1 mg/L, 2.3 mg/L, 3.1 mg/L, 4.9 mg/L, 5.1 mg/L, 5.0 mg/L and 19.6 mg/L, the saturated adsorption of the eight SAs were 6.0 mg/m, 8.0 mg/m, 8.8 mg/m, 12.0 mg/m, 19.2 mg/m, 20.0 mg/m, 19.6 mg/m, and 67.2 mg/m for the MOF-polymer monolithic column, respectively.

To evaluate the adsorption process of the MOF-polymer monolithic column for the eight SAs, static adsorption data were fitted with Langmuir and Freundlich isotherm models, the results of which are shown in Appendix A, and listed in Table 1 with the values of the corresponding parameters for the isotherm models. Comparison of the R^2^ values from both isotherm models indicates that a better fit is obtained using the Freundlich equation.

#### 2.4.2. Extraction Performance of the MOF-Polymer Monolithic Column

In order to further investigate the extraction ability of the MOF-polymer monolithic column, the polymer monolithic column without incorporation of the MOF was used as well for comparison. The extraction performance of both monolithic columns was studied using 50.0 µg/L SAs. Figure 5B indicates that the MOF-polymer monolithic column possesses a higher extraction capacity than the polymer monolithic column. Additionally, the enrichment factor of MOF-polymer monolithic column for eight SAs was calculated by comparing the peak areas obtained with MOF-polymer monolithic column extraction and without preconcentration. The enrichment factors were in the range of 42–56 for eight SAs.

Reproducibility and stability of the MOF-polymer monolithic column are crucial parameters for SPME. In this sense, a column-to-column reproducibility study was performed and estimated by calculating the relative standard deviations (RSDs) for extraction of the eight SAs. Satisfactory RSDs were obtained, ranging from 0.9 to 3.4% for intra-batches and 3.4 to 11.1% for inter-batches, revealing that the MOF-polymer monolithic column could be reused at least 100 times without any obvious loss of adsorption capacity. Similarly, this method is reproducible for both intra-day and inter-day studies, with RSDs < 10% for each. These results illustrate that the developed method achieves high reproducibility and satisfactory reliability for the detection of SAs, providing enhanced extraction performance when using the MOF-polymer monolithic column.

### 2.5. Application of the MOF-Polymer Monolithic Column

#### 2.5.1. Analytical Method Validation

Using the optimized SPME conditions described above, a method for the analysis of eight SAs using the MOF-polymer monolithic column for extraction coupled with UHPLC-MS/MS detection was developed. To validate the proposed method, the regression equations, linear ranges, linearity coefficients (R^2^), limits of detection (LODs), and RSDs were assessed. The results are summarized in Table 2, with linear ranges between 0.015 and 25.0 µg/L for all eight SAs. The R^2^ values range from 0.9954 to 0.9998 and the LODs are between 1.3 and 4.7 ng/L, with RSDs in the range of 2.1 to 6.1%.

#### 2.5.2. Sample Analysis

In our experiment, a new established analytical method based on NH_2_-MIL-53(Al) incorporating poly (AAPBA/MAA-co-EGDMA) was developed further herein to determine the residue of SAs present in liquid extracts from chicken and fish. The results are summarized in Table 3. Most of the SAs could be detected in both samples using this method. Likewise, spiked recoveries were conducted for spiked chicken and fish samples with SAs at concentration levels of 0.5 and 5.0 µg/Kg upon the addition of standard solutions. The typical chromatogram is shown in Figure 6, the spiked recoveries from chicken and fish samples were 85.7 to 115% and 86.7 to 113%, with corresponding RSDs between 2.5 and 7.3% and 2.0 and 6.6%, respectively. The above-mentioned results illustrate that this sensitive method is reliable, highly accurate, and practical for the analysis of trace SAs from real samples.

## 3. Materials and Methods

### 3.1. Chemicals and Reagents

Sulfadiazine (SDZ), sulfathiazole (STZ), sulfamerazine (SMI), sulfamethazine (SMZ), sulfamonomethoxine (SMM), sulfamethoxazole (SMX), sulfisoxazole (SIZ), sulfadimethoxine (SDM), internal standard sulfamethazine-D_4_, ethylene glycol dimethacrylate (EDGMA), methacrylic acid (MAA), 3-acrylamidophenylboronic acid (AAPBA), and NH_2_-BDC were all purchased from J&K Scientific (Beijing, China). Azo(bis)-isobutyronitrile (AIBN) was purchased from Aladdin Biochemical Technology Co., Ltd. (Shanghai, China). Methanol (MeOH) and acetonitrile (ACN) were purchased from Dikma (Beijing, China). Also, 3-(Trimethoxysilyl) propyl methacrylate (KH-570) and acetic acid (CH_3_COOH) were produced in the Tianjin Fuchen Chemical Reagent Factory. All reagents were analytical grade unless otherwise noted. The ultra-pure water used in all experiments was obtained from a Milli-Q gradient A10 system (Millipore, UK).

### 3.2. Instrumentation

A Shimadzu UHPLC-MS/MS 8050 system (Shimadzu, Japan) coupled with an electrospray ionization (ESI) source was used to analyze the SAs recoveries. The surface morphologies of the MOF-polymer monolithic column were observed using scanning electron microscopy (SEM), which was carried out on a SU8020 SEM instrument (Hitachi, Japan). X-ray diffractometry (XRD) measurements were conducted on a Bruker AXS D8 ADVANCE diffractometer (Karlsruhe, Germany). Fourier-transform infrared (FT-IR) spectroscopy using a Nicolet iS10 FT-IR spectrophotometer (Thermo Fisher Scientific, Madison, WI, USA) scanned between 4000 cm^−1^ and 500 cm^−1^ was employed to obtain FT-IR spectra. Thermogravimetric analysis (TGA) was performed on a NETZSCH STA 449 F3 Jupiter instrument (Netzsch, Selb, Germany). X-ray photoelectron spectroscopy (XPS) was obtained on an ESCALAB 250XI spectrometer (Thermo Fisher Scientific, Madison, WI, USA), which was used to detect the contents of the elements and chemical states. Brunauer-Emmett-Teller (BET) surface areas and porosimetry analyses were calculated from N_2_ adsorption measurements using an ASAP 2460 instrument (Atlanta, GA, USA).

### 3.3. Chromatographic and Spectrometric Conditions

The isocratic reversed-phase separation of analytes was carried out on a Shim-pack XR-ODS-Ⅲ (2.0 × 150 mm i.d., 2 μm) from Shimadzu. The mobile phase consisted of acetonitrile/water containing 0.5% acetic acid (*v*/*v*) at a constant flow rate of 0.3500 mL/min. The gradient elution program involved an increase of the acetonitrile concentration from 10% to 40% during 10 min. The column temperature was maintained at 36 °C. The sample injection volume was 10.000 μL. For MS analysis, quantitative analysis was performed using the multiple-reaction monitoring (MRM) mode. The analysis of SAs was performed in positive ion model with a 4.5 kV capillary voltage. Nebulizer gas and drying gas flow rates of 3.0 L/min and 15.0 L/min, respectively, were employed. The optimized parameters for ESI-MS, including the Q1 Pre Bias (V), Q3 Pre Bias (V), and collision energy (CE) for product ions are listed in Appendix A. Total ion chromatography of SAs are shown in Appendix A.

### 3.4. Preparation of NH_2_-MIL-53(Al)

NH_2_-MIL-53(Al) (MIL=Materials of Institute Lavoisier) was prepared with slight modification according to the literature procedure of Ahnfeldt et al. [33]. Briefly, 3.756 g NH_2_-BDC, 4.935 g AlCl_3_·6H_2_O, and 50 mL ultra-pure water were added to a 100 mL stainless steel reactor lined with Teflon. The reactor was then sealed and placed in an oven at 150 °C for 20 h. The obtained product was then activated with DMF at 150 °C for 4 h. Residual NH_2_-BDC was then removed, and the resulting activated NH_2_-MIL-53(Al) was dried in a vacuum oven. Subsequent characterization using XRD, FT-IR spectroscopy, and XPS was performed.

### 3.5. MOF-Polymer Monolithic Column Preparation

Firstly, fused-silica capillaries (4 m × I.D. 530 μm) were connected on the six-port valve and injected with 1 mol/L NaOH and 1 mol/L HCl by liquid pump, and then immersed for 4 h before being rinsed with purified methanol and water, followed by drying at 150 °C for 2 h. The activated capillaries were then filled with a KH570/methanol (1:1, *v*/*v*) solution. The reaction proceeded at ambient temperature for 6 h, and then the capillaries were rinsed with methanol, dried with nitrogen, and sealed with rubber stoppers at each end. The activated capillaries were then cut into pieces 15 cm in length. Secondly, NH_2_-MIL-53(AL) (24.0 mg), functional monomer AAPBA (11.3 mg), and the initiator AIBN (4.5 mg) were mixed in DMSO (250 μL). The functional monomer MAA (19.2 μL) and porogenic solvents toluene (580 μL) and isooctane (250 mL) were then added, followed by the EGDMA (175 μL) cross-linker. The mixture was then homogenized and degassed in an ultrasonic bath for 5 min and thereafter siphoned to the pretreated capillary column. The capillary was finally sealed with silicone rubber at each end and then placed in a 60 ± 1.0 °C oven to initiate the polymerization reaction for 48 h. The capillary column was then dried at 110 °C for 2 h for subsequent use. The obtained MOF-polymer monolithic column was synthesized by in situ technique and washed with methanol/acetic acid (9:1, *v*/*v*) to remove any unreacted reagents until they could no longer be detected with UHPLC-MS/MS. The monolithic capillary column was finally cut with a knife for a final length of 10.0 cm. For comparison, the control polymer monolithic column without MOF was also synthesized as described above, omitting the MOFs [34].

### 3.6. Adsorption Performance of the MOF-Polymer Monolithic Column

To investigate the adsorption performance of the MOF-polymer monolithic column, the solutions were diluted to different concentrations using a pH = 5.0 dilute HCl solution. In the static adsorption experiment, 5.0 mL of SAs standard solutions (V, L) with different concentrations (*C_i_*, mg/mL) were injected through the MOF-polymer monolithic column. After adsorption, the equilibrium adsorption capacity *Q_e_* (mg/m) was calculated from Equation (1) [32,35].
(1)$$Q_{e}\;=\;\frac{(C_{i}-C_{e})V}{L}$$
where *L* (m) is the length of the MOF-polymer monolithic column, and the equilibrium concentration of SAs (*C_e_*, μg/mL) was obtained experimentally.

In order to determine the optimal model to accurately depict the adsorption processes and evaluate the binding properties of the MOF-polymer monolithic column, the adsorption isotherm data was fitted with the most commonly used models of Langmuir (Equation (2)) and Freundlich (Equation (3)) isotherms [32,35,36,37].
(2)$$Q_{e}=\frac{\;Q_{\max}K_{L}C_{e}}{1+C_{e}}$$
(3)$$LgQ_{e}=Lg(\frac{C_{e}}{n}\times K_{F})$$
where *Q_e_* (ng) and *C_e_* (mg/L) are the experimentally measured equilibrium adsorption capacity and the equilibrium concentration. *Q_max_* (ng), *K_L_*, and *K_F_* are the maximum amounts of adsorbate, the Langmuir constant, and the Freundlich constant, respectively, which represent the capacity for adsorption obtained by the appropriate fitting model, and *n* is the heterogeneity factor.

### 3.7. Sample Preparation

Live chicken and fish were purchased from a local market. Chickens (0.6 kg) were fed with corn containing 1500 μg/Kg SAs and fish (0.6 kg) were fed in water containing 1500 μg/Kg SAs for one week before being slaughtered. Samples obtained from these animals were homogenized and stored at –20 °C before use. First, a 1.00 g sample was added to a 50 mL centrifuge tube. After addition of 1.0 g Na_2_SO_4_ and 10 mL acetonitrile, the mixture was extracted by sonication for 10 min and the upper liquid was collected. This process was repeated in triplicate. The separate mixtures were then centrifuged at 12,000 r/min for 5 min. Afterward, the supernatants were collected and combined in the same centrifuge tube, and then 10 μL of 10.3% potassium ferrocyanide (*w*/*v*) and 10 μL of 21.9% zinc acetate (*w*/*v*) were added to the samples to remove the protein and centrifuged for 5 min at 12000 rpm [38]. The solution was then added to 10.0 mL of hexane, oscillation, and the supernatant was removed. The extraction process was repeated. The acetonitrile solution was dried using reduced pressure distillation at 40 °C. The residue was then dissolved with 10.0 mL HCl solution (pH = 5.0) and passed through a 0.22 µm nylon filter. Finally, the sample solution was enriched using the MOF-polymer monolithic column.

## 4. Conclusions

In this study, a novel method to prepare a MOF-polymer monolithic column was performed by adding NH_2_-MIL-53(Al) to a polymer monolithic column, which was successfully fabricated and characterized. The NH_2_-MIL-53(Al) support contained pores of an appropriate size to fit the polymer monolithic column, while remaining compact and improving the adsorption performance. Therefore, the MOF-polymer monolithic column largely enhanced the extraction performance, water stability, and specific surface areas, whereas the polymer column that did not contain a MOF showed limited performance for these parameters. Optimized conditions were established to pursue this as a feasible analytical method for the detection of SAs. The MOF-polymer column was successfully used to determine eight SAs in chicken and fish samples by applying it in SPME coupled with UHPLC-MS/MS. The detection of SAs indicates that this method provides good selectivity and high recoveries and precisions. The detection of trace SAs from chicken and fish samples by using this newly-developed method was satisfactory, making this method a promising approach for the detection of trace amounts of analytes from the complex matrices present in foods.

## Figures and Tables

**Figure 1 molecules-25-00897-f001:**
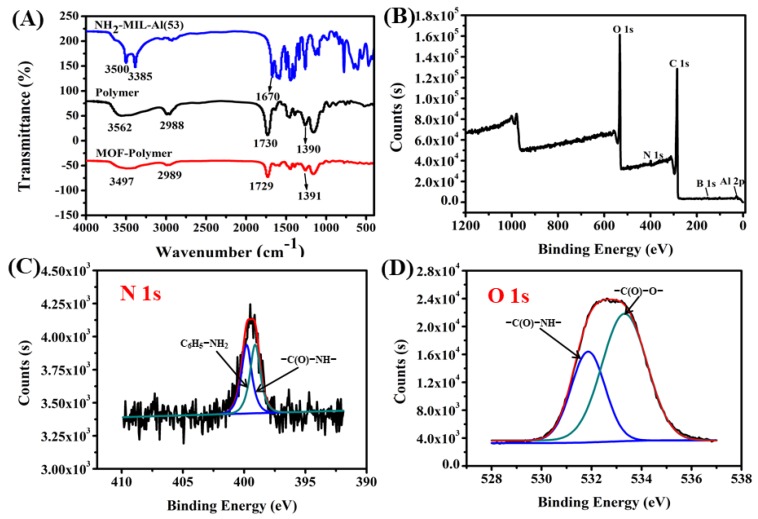
(**A**) FT-IR spectra of NH_2_-MIL-53(Al), polymer, and metal–organic framework (MOF)-polymer; (**B**) full scan XPS spectrum of the MOF-polymer; and high-resolution XPS spectra of the MOF-polymer (**C**) N 1s and (**D**) O 1s showing deconvolution of the peaks.

**Figure 2 molecules-25-00897-f002:**
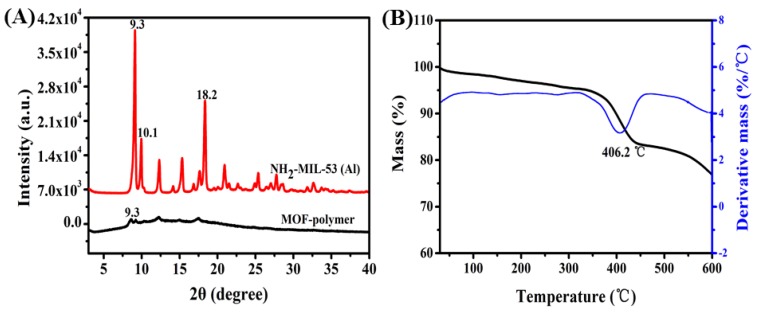
(**A**) XRD patterns of NH_2_-MIL-53(Al) and the MOF-polymer and (**B**) thermogravimetric curves (black line) and derivative curves (blue line) of the MOF-polymer.

**Figure 3 molecules-25-00897-f003:**
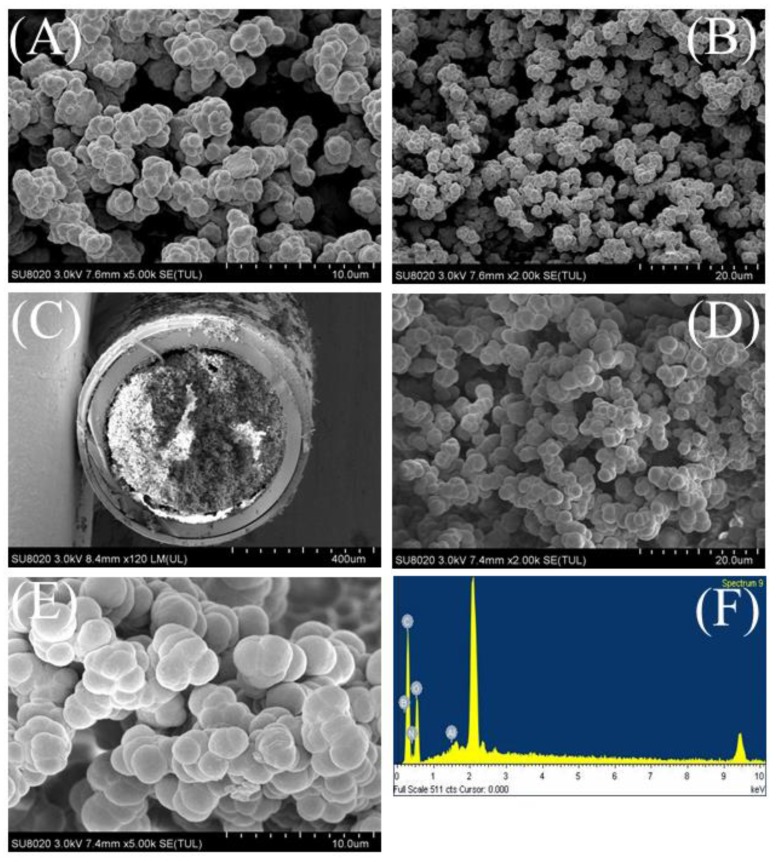
(**A** and **B**) SEM micrographs of the polymer monolith column. (**C**–**E**) SEM micrographs and (**F**) EDS spectrum of the MOF-polymer monolithic column. Magnifications of SEM images are: (**A**) 5000×, (**B**) 2000×, (**C**) 120×, (**D**) 3000×, and (**E**) 5000×.

**Figure 4 molecules-25-00897-f004:**
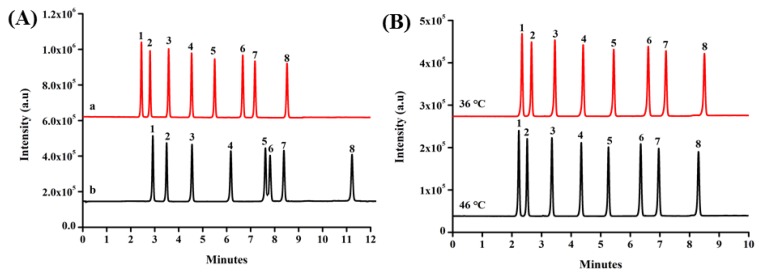
(**A**) The total ion chromatography of different mobile phases. (a) Acetonitrile/water containing 0.5% acetic acid (*v*/*v*); (b) methanol/water containing 0.5% acid (*v*/*v*); (**B**) the influence of the column temperature. 1: SDZ, 2: STZ, 3: SMI, 4: SMZ, 5: SMM, 6: SMX, 7: SIZ, 8: SDM.

**Figure 5 molecules-25-00897-f005:**
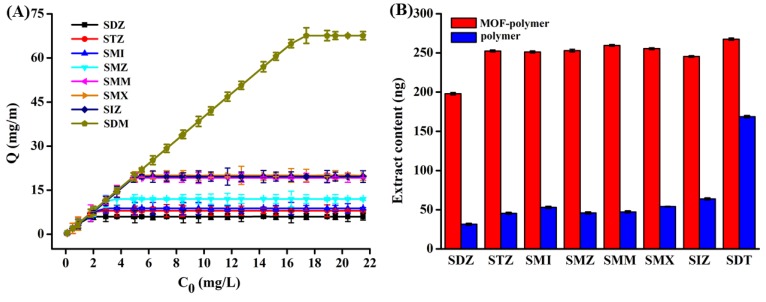
(**A**) Adsorption isotherms on MOF-polymer monolithic column of the eight SAs used in this study. A 10 cm MOF-polymer monolithic column was used to adsorb different concentrations of SAs from 5.0 mL of aqueous solution at pH = 5.0. (**B**) Comparison of the amounts of extracted SAs using the MOF-polymer monolithic and polymer monolithic columns.

**Figure 6 molecules-25-00897-f006:**
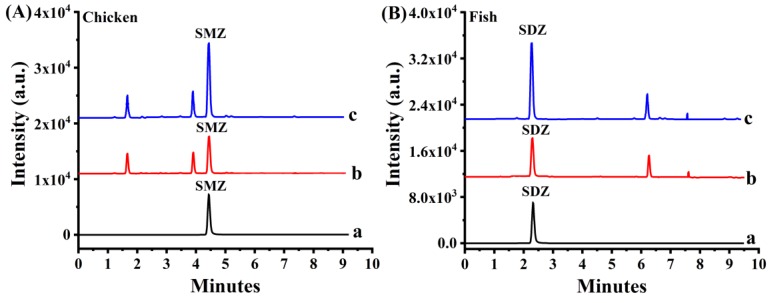
Typical chromatograms for SMZ in chicken samples (**A**), and SDZ in fish samples (**B**) using in-tube SPME-UHPLC−MS/MS analysis: (a) a direction injection standard solution at 10.0 µg/L; (b) the extract of sample; (c) the spiked sample with 0.50 µg/kg of each analyte.

**Table 1 molecules-25-00897-t001:** Adsorption isotherm parameters of SAs on the MOF-polymer monolithic column.

Analytes	Langmuir				Freundlich	
	*Q*_max_ (mg/m)	K_L_ (L/mg)	R^2^	K_F_ (L/mg)	n	R^2^
SDZ	334.22	5.05	0.80306	11.63	0.55	0.88174
STZ	1284.03	5.14	0.76070	9.05	0.70	0.91181
SMI	874.13	5.55	0.69402	25.63	0.64	0.84180
SMZ	366.57	6.58	0.71091	17.06	0.52	0.88569
SMM	520.29	15.64	0.90646	26.44	0.69	0.98678
SMX	78.31	20.66	0.78464	59.14	0.40	0.82074
SIZ	86.36	10.62	0.72585	31.42	0.40	0.79187
SDM	355.87	66.23	0.95141	114.11	0.67	0.99618

**Table 2 molecules-25-00897-t002:** Calibration curve equations, linear ranges, detection limits, and precision of the MOF-polymer-monolithic-column-based SPME coupled to UHPLC-MS/MS as a detection method for extraction and assessment of SAs from aqueous solution at pH = 5 (*n* = 5).

Analytes	Regression Equation	R^2^	Linear Range (µg/L)	LOD ^a^ (ng/L)	LOQ ^a^ (ng/L)	RSD ^b^ (%)
**SDZ**	y = 0.2911x − 0.000883	0.9998	0.015–25.0	4.7	14.2	4.7
**STZ**	y = 0.3201x + 0.0243	0.9954	0.015–25.0	3.1	9.35	4.4
**SMI**	y = 0.3595x + 0.0303	0.9989	0.015–25.0	1.7	5.30	6.1
**SMZ**	y = 0.3441x + 0.0207	0.9997	0.015–25.0	2.4	7.42	4.8
**SMM**	y = 0.3448x − 0.00505	0.9994	0.015–25.0	3.5	10.9	4.2
**SMX**	y = 0.3666x + 0.00663	0.9967	0.015–25.0	1.3	4.15	3.7
**SIZ**	y = 0.3617x + 0.0395	0.9972	0.015–25.0	2.3	6.96	2.2
**SDM**	y = 0.3137x + 0.0167	0.9996	0.015–25.0	3.8	11.7	2.1

^a^ Detection of limits and quantification of limits were estimated on the basis of 3:1 and 10:1 signal-to-noise ratios, respectively. ^b^ Relative standard deviation (RSD) was monitored using 5.0 µg/L sulfonamides mixed solution.

**Table 3 molecules-25-00897-t003:** The MOF-polymer monolithic column SPME-UHPLC-MS/MS analysis of eight trace SAs in chicken and fish samples.

Sample	Analytes	Concentration (µg/Kg)	RSD (%)	Spiked Concentration (µg/Kg)			
				0.50 (µg/Kg)		5.0 (µg/Kg)	
				Recovery (%)	RSD (%)	Recovery (%)	RSD (%)
Chicken	SDZ	0.0951	9.3	85.7	7.3	92.7	2.5
	STZ	0.0507	8.3	115	4.2	103	3.6
	SMI	0.747	6.2	110	6.0	107	5.2
	SMZ	0.620	7.4	108	5.4	93.6	3.8
	SMM	0.145	7.2	98.8	2.7	107	4.1
	SMX	N.Q.	--	91.9	5.5	97.3	2.8
	SIZ	0.147	6.5	88.1	4.3	96.1	4.6
	SDM	1.81	3.9	112	3.9	104	5.7
Fish	SDZ	0.494	7.8	96.8	6.6	101	4.7
	STZ	0.228	5.6	89.9	5.6	105	3.6
	SMI	3.09	6.2	95.8	5.1	95.3	5.1
	SMZ	4.28	4.8	89.0	4.0	96.0	3.1
	SMM	1.23	7.2	86.7	4.3	96.6	3.8
	SMX	0.554	6.2	86.8	5.2	89.5	2.0
	SIZ	1.22	8.3	102	6.3	113	6.2
	SDM	7.89	4.0	104	3.9	85.6	3.2

N.Q. = not quantified.

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
