# Peer review of "NH2-MIL-53(Al) Polymer Monolithic Column for In-Tube Solid-Phase Microextraction Combined with UHPLC-MS/MS for Detection of Trace Sulfonamides in Food Samples"

_molecules, 2020, doi:10.3390/molecules25040897_

Round 1
Reviewer 1 Report
Comments:
This paper describes about a novel porous monolithic capillary column prepared via in situ polymerization for solid-phase microextraction (SPME) to efficiently adsorb trace sulfonamides from food samples. This paper is interesting and optimized SPME conditions well, but the following points should be considered.
1. Extraction yields or enrichment factors of sulfonamides by SPME with the proposed monolithic capillary column should be given in the text.
2. Typical chromatograms obtained from standard solution and food samples should be shown in figure.
3. Sample preparation of food samples is complicate and time-consuming. Please indicate the analysis time required from sampling to data analysis per sample.
Author Response
Firstly, we are very grateful to editors and reviewers for the comments which would be valuable for the improvement of our paper. We have carefully considered the comments and have revised the manuscript accordingly. All the additions and modifications have been highlighted in red in the revised manuscript. Response to reviewer’s comments was explained point-to-point as following. Thank you for handling this paper again!
Extraction yields or enrichment factors of sulfonamides by SPME with the proposed monolithic capillary column should be given in the text.
Response: Thanks for your comments on our paper. The enrichment factors were measured from 42 to 56 for eight sulfonamides, which were specified in lines 200-202 in the revised manuscript.
Typical chromatograms obtained from standard solution and food samples should be shown in figure.
Response: Thanks to the valuable suggestion. The typical chromatograms have been added into the revised manuscript. (see Figure 5 and Figure 5S in the revised manuscript).
Sample preparation of food samples is complicate and time-consuming. Please indicate the analysis time required from sampling to data analysis per sample.
Response: It is an interesting concern. Sample preparation of food samples is indeed complicate and time-consuming, which spent about 180 min from sampling to data analysis per sample
Reviewer 2 Report
Overall, manuscript is well presented and described. However, following comments are recommended before accepting the manuscript.
Line 15: What is MIL? Please define.
For optimization experiments please provide all the parameters which kept constant. Ex: at which concentration of SAs authors did the pH optimization?
Table 2: Please provide LOQs as well. LOD is only good for identification/confirmation.
Table 3: Spiked recovery units are inconsistent, ug/L or ug/Kg. Please clarify? What is the enrichment factor during the detection?
Author Response
Response to reviewer’s comments
Firstly, we are very grateful to editors and reviewers for the comments which would be valuable for the improvement of our paper. We have carefully considered the comments and have revised the manuscript accordingly. All the additions and modifications have been highlighted in red in the revised manuscript. Response to reviewer’s comments was explained point-to-point as following. Thank you for handling this paper again!
Line 15: What is MIL? Please define.
Response: The reviewer raises an interesting concern. We are very sorry for the unclear expression in the manuscript. Reference 33 report that MIL (Materials of Institute Lavoisier). The related content has been revised to make the expression in the revised manuscript.
For optimization experiments please provide all the parameters which kept constant. Ex: at which concentration of SAs authors did the pH optimization?
Response: Thanks for your comments on our paper. For optimization experiments, the kept constant parameters were 5.00 mL of 50.0 µg/L standard solutions of the eight different SAs. In our paper, the pH optimization was investigated using 50.0 µg/L standard solutions of the eight different SAs at pH= 5.0 (see lines 143-146 in the revised manuscript).
Table 2: Please provide LOQs as well. LOD is only good for identification/confirmation.
Response: Thanks for your comments on our paper. LOQ was estimated on the basis of 10:1 signal to noise ratios. We have revised our paper according to your suggestion.
Table 3: Spiked recovery units are inconsistent, ug/L or ug/Kg. Please clarify? What is the enrichment factor during the detection?
Response: We very much appreciate for the careful reading of our manuscript. The units mistakes listed above have been corrected. Besides these changes, we have carefully checked the manuscript for other problems and corrected them. The enrichment factor of MOF-polymer monolithic column for eight SAs was calculated by comparing the peak areas obtained with MOF-polymer monolithic column extraction and without preconcentration. The enrichment factors were in range of 42-56 for eight SAs.
Reviewer 3 Report
The manuscript reports a study dealing with the development of a MOF-polymer sorbent for the in-tube solid phase microextraction of sulfonamides in liquid samples followed by analysis carried out by UHPLC-MS/MS.
The newly introduced sorbent shows interesting properties and the proposed approach seems suitable for the analysis of sulfonamides; the material was deeply characterized, the extraction protocol was carefully optimized and the extraction performance are adequate.
However, some points need to be re-elaborated and clarified. In particular:
The title should be revised because it should be specified that the technique is based on an in-tube SPME, furthermore it should be highlighted that the samples, if solid, require a prior liquid extraction with an adequate solvent. These aspects should emerge also in the abstract Please, write UHPLC instead of UPLC because the name UPLC is a trademark of Waters (https://trademarks.justia.com/783/52/uplc-78352811.html) and the authors used a Shimadzu instrument Although the authors emphasize the advantages of UHPLC analysis compared to other chromatographic techniques, any UHPLC data or chromatogram is reported; a paragraph concerning the analysis step should be added in the results and discussion section Line 55: please specify that the polymer is incorporated “inside” the capillary Line 59: do the authors mean “adequate” instead of “inadequate”? Line 77: the extraction efficiency of the MOF-polymer is increased compared to what? Have the authors compared the proposed device to commercial available fibres? Lines 84-85: please chemically define the MOF and the polymer the first time that they are cited in the manuscript Lines 131-132: how was surface area measured? Caption to figure 3: is really the magnification of figures 3C 3000×? Line 141: “optimal extraction conditions” is more correct than “enrichment conditions” Paragraphs 2.2.1 and 2.2.2: please report the fixed conditions and the volume and the solvent in which the sample is diluted Line 159: please report the solvent used to clean the capillary Paragraph 2.2.4: why a higher volume was selected? If the recovered analyte absolute amount is the same a higher volume results in a higher dilution and therefore a higher limit of detection/quantification Paragraph 3.4.2: it should be highlighted that the analytes have to be pre-extracted from a solid sample with and adequate solvent Paragraph 3.3: it is not clear if the analyses are carried out in isocratic or gradient mode. In the second case, the time needed for the gradient should be reported Paragraph 3.5: please, add a reference for the method of preparation of the polymer Line 293: how is the capillary filled? Paragraph 3.7: is the extraction method a literature method? If yes, please report the reference
Author Response
Response to reviewer’s comments
Firstly, we are very grateful to editors and reviewers for the comments which would be valuable for the improvement of our paper. We have carefully considered the comments and have revised the manuscript accordingly. All the additions and modifications have been highlighted in red in the revised manuscript. Response to reviewer’s comments was explained point-to-point as following. Thank you for handling this paper again!
The title should be revised because it should be specified that the technique is based on an in-tube SPME, furthermore it should be highlighted that the samples, if solid, require a prior liquid extraction with an adequate solvent.
Response: Thank you for the valuable suggestion. We have revised our paper according to your suggestion.
These aspects should emerge also in the abstract Please, write UHPLC instead of UPLC because the name UPLC is a trademark of Waters (https://trademarks.justia.com/783/52/uplc-78352811.html) and the authors used a Shimadzu instrument Although the authors emphasize the advantages of UHPLC analysis compared to other chromatographic techniques, any UHPLC data or chromatogram is reported;
Response: Thanks for your comments on our paper. We have revised our paper according to your suggestion.
a paragraph concerning the analysis step should be added in the results and discussion section
Response: Thank you for the suggestion. We have revised our paper according to your suggestion.
Line 55: please specify that the polymer is incorporated “inside” the capillary
Response: Thank you for the suggestion. MOF-polymer monolithic column was synthesized by in situ technique. We have revised our paper according to your suggestion.
Line 59: do the authors mean “adequate” instead of “inadequate”?
Response: We very much appreciate for the careful reading of our manuscript. We have revised our paper according to your suggestion.
Line 77: the extraction efficiency of the MOF-polymer is increased compared to what? Have the authors compared the proposed device to commercial available fibres?
Response: Thanks for your comments on our paper. Compared with the polymer monolithic column, the MOF-polymer monolithic column noticeably improves the extraction efficiencies of SAs. In addition, commercial fiber, such as PDMS fiber, PDMS/DVB fiber, PA fiber, CW/DVB fiber and CAR/PDMS fiber are certain thickness (50-100 µm) and length (10 mm), prepared MOF-polymer in-tube solid-phase microextraction column (10.0 cm × O.D. 530 μm),therefore,there are not investigated to comparison of prepared MOF-polymer in-tube solid-phase microextraction column with commercial fiber.
Lines 84-85: please chemically define the MOF and the polymer the first time that they are cited in the manuscript
Response: Thank you for the suggestion. We have revised our paper according to your suggestion. (see lines 16 and 62 in the revised manuscript).
Lines 131-132: how was surface area measured? Caption to figure 3: is really the magnification of figures 3C 3000×?
Response: We very much appreciate for the careful reading of our manuscript. In our paper, the surface area is measured by Brunauer-Emmett-Teller. The figures 3C 120× is right, we have revised our paper.
Line 141: “optimal extraction conditions” is more correct than “enrichment conditions”
Response: Thank you for the valuable suggestion. We have revised our paper according to your suggestion.
Paragraphs 2.2.1 and 2.2.2: please report the fixed conditions and the volume and the solvent in which the sample is diluted
Response: Thank you for the suggestion. These conditions were investigated using 5.00 mL of 50.0 µg/L standard solutions of the eight different SAs in ultra-pure water. We have revised our paper according to your suggestion.
Line 159: please report the solvent used to clean the capillary
Response: Thank you for the suggestion. The ultra-pure water was used to clean the column. We have revised our paper according to your suggestion.
Paragraph 2.2.4: why a higher volume was selected? If the recovered analyte absolute amount is the same a higher volume results in a higher dilution and therefore a higher limit of detection/quantification
Response: Thank you for the suggestion. We are very sorry for the unclear expression in the manuscript. The SDT extracted in the MOF-polymer monolithic column can be almost completely transferred by 400 µL of eluent, the eluent was concentrated to dry under nitrogen and the residue was dissolved with 100 µL of acetonitrile/water (2:8, v/v). Finally, 10 μL of elution was injected for UHPLC-MS/MS analysis.
Paragraph 2.4.2: it should be highlighted that the analytes have to be pre-extracted from a solid sample with and adequate solvent
Response: Thank you for the suggestion. it should be highlighted that the analytes have to be pre-extracted from a solid sample with and adequate solvent in paragraph 3.7.
Paragraph 3.3: it is not clear if the analyses are carried out in isocratic or gradient mode. In the second case, the time needed for the gradient should be reported
Response: Thank you for the suggestion. The gradient elution program involved an increase of the acetonitrile concentration from 10% to 40% during 10 min.
Paragraph 3.5: please, add a reference for the method of preparation of the polymer
Response: Thank you for the suggestion. We have added a reference 37 for the method of preparation of the polymer our paper according to your suggestion.
Line 293: how is the capillary filled?
Response: Thanks for your comments on our paper. Fused-silica capillaries were connected on the six-port valve and injected with 1 mol/L NaOH and 1 mol/L HCl by liquid pump. We have revised our paper according to your suggestion.
Paragraph 3.7: is the extraction method a literature method? If yes, please report the reference
Response: Thanks for your comments on our paper. The extraction method is empirical method and our previous work. We have revised our paper according to your suggestion.
Round 2
Reviewer 3 Report
The Authors answered to the reviewer’s remarks and the overall quality of the manuscript therefore increased making the article now suitable for the publication in Molecules.
However, some points should be further clarified revised:
The manuscript n should reflect that the samples, if solid, require a prior liquid extraction with an adequate solvent. I propose the following modifications: line 30 “…monitoring trace sulfonamides in food samples previously extracted with an adequate solvent.”, line 75 “…while the combination of liquid extraction, MOF polymer-monolithic-column-based SPME….”, line 239 “…to determine the residue of SAs present in liquid extracts from chicken and fish.” My comment on the addition of a paragraph concerning the analysis step has been misunderstood: I suggested to add a paragraph on the optimization of the analysis (i.e. chromatographic) conditions and not a paragraph on the optimized sampling condition. Paragraph 2.4.1 is therefore redundant and I suggest to remove it. Concerning analysis, the introduction of the two figures containing the chromatograms in my opinion can be enoughAuthor Response
Response to reviewer’s comments
Firstly, we are very grateful to reviewer and editors for the comments which would be valuable for the improvement of our paper. We have carefully considered the comments and have revised the manuscript accordingly. All the additions and modifications have been highlighted in red in the revised manuscript. Response to reviewer’s comments was explained point-to-point as following. Thank you for handling this paper again!
- line 30 “…monitoring trace sulfonamides in food samples previously extracted with an adequate solvent.”,
Response: Thank you for the valuable suggestion. We have revised our paper according to your suggestion.
- line 75 “…while the combination of liquid extraction, MOF polymer-monolithic-column-based SPME….”,
Response: Thanks to your valuable suggestion. We have revised our paper according to your suggestion.
- line 239 “…to determine the residue of SAs present in liquid extracts from chicken and fish.”
Response: Thank you for the valuable suggestion. We have revised our paper according to your suggestion.
- My comment on the addition of a paragraph concerning the analysis step has been misunderstood: I suggested to add a paragraph on the optimization of the analysis (i.e. chromatographic) conditions and not a paragraph on the optimized sampling condition. Paragraph 2.4.1 is therefore redundant and I suggest to remove it. Concerning analysis, the introduction of the two figures containing the chromatograms in my opinion can be enough
Response: Thank you for the valuable suggestion. We have revised our paper according to your suggestion.